# Status and associated factors of psychological resilience of Chinese medical aid team members under public health emergencies in a cross-sectional study

Xueyi Wei[1], Yinhua Liang[2]*, Jun Huang[2]

**1** Xiangya School of Nursing, Central South University, Changsha, China, **2** Teaching and Research Section of Clinical Nursing, Xiangya Hospital, Central South University, Changsha, China

* liangyh@csu.edu.cn

## Abstract

### Background

Chinese medical aid team members (CMATMs) play an important role in the implementation of China's foreign health aid strategy. However, there is no research on the status and influencing factors of their psychological resilience.

### Objective

This study aims to investigate the current status and influencing factors of psychological resilience of CMATMs.

### Methods

From 3rd December, 2024–25th December, 2024, a convenience sampling method was used. Demographic information questionnaire, Connor Davidson Resilience Scale, General Self-Efficacy Scale, Self-Rating Anxiety Scale and Self-Rating Depression Scale were used to measure the psychological status of aid members. The influencing factors of psychological resilience were investigated through *T*-test, ANOVA, pearson correlation analysis, Kruskal-Wallis test and multiple linear regression.

### Results

A total of 89 questionnaires were analysed. The total score of psychological resilience was $72.53 \pm 13.48$, which was generally at a good level. Pearson correlation analysis showed that self-efficacy was positively correlated with psychological resilience, while anxiety and depression were negatively correlated with psychological resilience. Multiple linear regression analysis revealed that factors related to psychological resilience included professional title($B = 4.679$, $P = 0.001$), loneliness($B = 4.973$,

**Data availability statement:** All relevant data are within the paper and its Supporting Information files.

**Funding:** The study was supported by the National Natural Science Foundation of China [82160554], Natural Science Foundation of Hunan Province [2020JJ8051] and Project Program of National Clinical Research Center for Geriatric Disorders [2021LNJJ15]. The funder mainly supported us in data collection. Furthermore, should there be any publication fees, the funder will kindly assist us in covering those costs.

**Competing interests:** The authors have declared that no competing interests exist.

$P = 0.000$), classification of hardship areas in recipient countries($B = 1.864$, $P = 0.004$), family support($B = 3.110$, $P = 0.001$), cultural and entertainment($B = 3.537$, $P = 0.001$), and direct participation in public health emergency response missions in recipient countries($B = -3.775$, $P = 0.006$).

## Conclusions

In the context of public health emergencies, CMATMs are prone to great pressure, which has a negative impact on their psychological resilience. This study provides a theoretical basis for formulating psychological intervention measures for CMATMs. Policymakers or healthcare administrators should regular assess CMATMs ' mental health status, strengthen mental health training, mobilize family and social support networks, improve infrastructure and welfare security policies to help them establish a more adaptive defense mechanism and maintain good psychological resilience during foreign aid.

## 1 Introduction

In recent years, with the outbreak of Severe Acute Respiratory Syndrome (SARS), the global spread of H1N1, the epidemic of Ebola virus disease (EVD) in West Africa, the global pandemic of Corona Virus Disease 2019(COVID-19), and the frequent occurrence of various emergencies, the international community has entered a high-risk period of public health emergencies [1]. As the world's second largest economy and one of the largest emerging donors of foreign aid, China began to provide overseas aid to other socialist countries shortly after its independence in 1949 [2]. For decades, China has sought to participate in global governance, especially in the area of global health. Sending medical relief teams to recipient countries is one of the main ways to provide health assistance to them, and it is also one of the concrete manifestations of China's active participation in global governance [3,4]. Chinese Medical Aid Team is a team sent by China to provide free medical services in underdeveloped countries, and is an important part of China's foreign exchanges and cooperation [5]. According to the survey, the total amount of Chinese medical assistance projects is 5.67 billion US dollars [6], a total of 26,000 medical members have been dispatched, and thousands of CMATMs are still providing services in recipient countries [7].

Due to the distinct natural environment, social norms, religious culture, lifestyle and economic development level of different countries, CMATMs are prone to cross-culture stress during the transition from the familiar living environment to a foreign country. In addition, the destinations of them usually have high prevalence rates of infectious diseases such as HIV, malaria, dengue fever and Ebola. Facing high-risk infectious diseases and terrorist attacks, CMATMs, often isolated from their social support networks, are susceptible to a range of physiological and psychological stress reactions. In severe cases, this can impair their physical and mental health, disrupt their professional performance, and increase the risk of suicidal ideation [8,9].

One study have shown that 80% of aid volunteers experienced psychological problems, including loneliness (36%), helplessness (18%), irritability (12%), and depression (14%) [10]. In the context of public health emergencies, medical workers are more likely to face serious mental health difficulties [11]. Therefore, they need reasonable coping strategies to reduce the impact of environmental changes and daily life pressure on their mental health.

One of the strategies for effectively coping with stress and maintaining mental health is resilience. Resilience focuses on how individuals actively adapt to adversity and even grow from the experience, offering a more holistic perspective on mental well-being that goes beyond merely mitigating the negative impacts of stress [12,13]. This is particularly relevant to the long-term, high-intensity and cross-cultural working environment faced by CMATMs, which can help us understand the psychological adaptation process of medical aid workers more comprehensively and offer a broader protective framework with intervention potential. Psychological resilience refers to the resilience shown by an individual when experiencing negative events, trauma, changes and adversity [14]. It enables individuals to transform their self-destructive forces into a proactive outlook on life [15,16]. Resilience among health care workers is associated with increased job satisfaction and an improved quality of professional life. Furthermore, resilience plays a critical role in mitigating the risk of developing mental health challenges such as post-traumatic stress disorder, anxiety, and depression [17–19]. One study has shown that resilience can help nurses adapt to effectively withstand stressful situations in the context of the COVID-19 pandemic [12]. Cooper et al. [20] put forward the following six factors that affect resilience, including social support, self-efficacy, work-life balance, humor, optimism, and realism. Self-efficacy is a person's ability to perform at a predetermined level and to control the circumstances that affect their lives. As an internal protective factor, self-efficacy may reduce the adverse effects of stress [21]. Adequate social support is essential for effective management of stressful events, including emergencies, crises, and infectious disease outbreaks [21]. However, to our knowledge, no studies have investigated the factors affecting the resilience of CMATMs.

The stress-coping theory indicates that when individuals face stressful events, they will first conduct cognitive evaluations based on existing information and experience, and then respond to the stressful events, thereby solving problems or alleviating emotions [22]. During this process, an individual's inner responses, such as psychological resilience, play a significant role. Those with higher psychological resilience tend to adopt positive coping styles. Psychological resilience levels are a means of helping aid workers recover normal function and performance after experiencing a major public health emergency. Understanding their level of resilience can help assess their work performance and mental health. However, to date, little research has been done on this particular population, let alone its level of psychological resilience. This is a critical research gap given their pivotal role in implementing China's foreign health aid policy and their exposure to unique stressors in challenging environments. Stressors in one area can affect stress in another [21], therefore, identifying the factors that affect psychological resilience is crucial for coping with stressors and preventing negative emotions. This study aims to reveal the current status and influencing factors of psychological resilience of this population based on the stress-coping theory. Through the comprehensive analysis of these influencing factors, not only can provide a more accurate assessment of the psychological resilience of CMATMs, but also provide a scientific basis for follow-up intervention and support. Finally, it will help improve the overall adaptability and work stability of team members, ensure the smooth development of foreign medical aid work, promote international medical cooperation and exchanges, enhance the friendship and mutual trust between different countries.

## 2 Methods

### 2.1 Study design and participants

From 3rd December, 2024–25th December, 2024, a convenience sampling method was used. A total of 89 medical team members who carried out foreign aid missions from February 2023 to September 2024 were selected, and their psychological health status was investigated by sending questionnaires online form. The Guidelines for the Strengthening the Reporting of Observational Studies in Epidemiology(STROBE) were used [23].

 

For the sample size, the recommendations were between 5 and 10 times the number of independent variables [24]. With an estimated 34 independent variables (31 demographic variables, the total scores from General Self-Efficacy Scale, Self-Rating Anxiety Scale and Self-Rating Depression Scale), a minimum of 170 participants were required. The demographic variables with statistical significance in the univariate analysis were selected for multiple linear regression. Ultimately, 17 variables (14 demographic variables and the total scores of three scales) were included in the regression analysis. However, due to the special research population of CMATMs, the number was relatively small. In addition, team members were distributed in different countries and regions, it was difficult for researchers to contact all potential participants. Therefore, we did our best to select only 89 subjects. Relatively small sample size was undeniably a limitation of our research.

The inclusion criteria were as follows: (a) Members of the Chinese medical team for foreign aid, including medical and non-medical members. While medical professionals were the core of CMATMs, the non-medical members had an indirect impact on the emotional and mental health of the medical professionals. In order to fully understand the psychological resilience of the entire team, our study was not limited to the core medical staff, but also included non-medical staff; (b) informed consent and voluntary participation in the study. Exclusion criteria: (a) during the investigation period, the team members who successfully completed the foreign aid mission; (b) the duration of the foreign aid mission was less than 1 month, and (c)unable to participate due to physical illness.

## 2.2 Measurements

### 2.2.1 Demographic information questionnaire.
Demographic information consisted of two parts: general information about the members and their work and lifestyle in the recipient country. According to the purpose of the study, research team reviewed relevant literature and, considering the unique working and living conditions of medical aid team members, developed a demographic information questionnaire informed by stress-coping theory.

The general information questionnaire included 11 variables, namely age, sex, marital status, highest level of education education, occupation, level of the dispatching organization, professional title, prior overseas experience before participating in foreign aid work, classification of hardship areas in recipient countries, family support, and family fertility.

The second part included 20 variables, namely satisfaction with family life, satisfaction with work, smoking status, frequency of drinking, loneliness, culture and entertainment, frequency of exercise, average daily Internet use time, economic subsidy policy, professional title promotion policy, material condition, political situation, social security situation, language adaptation situation and climate adaptation situation of the recipient country, history of mosquito or insect bites, history of typhoid or malaria or COVID-19 infection, and direct participation in public health emergency response missions in recipient countries.

### 2.2.2 The Connor Davidson Resilience Scale(CD-RISC).
CD-RISC was developed by Connor and Davidson [25] and this study adopted the Chinese version translated by Yu et al. [26] based on Chinese culture and national conditions. The Chinese version of the scale had undergone cross-cultural debugging, confirmatory factor analysis and revision, modifying the five dimensions of the original scale to three dimensions.The scale includes three dimensions (25 items in total): optimism dimension (4 items), self-strengthening dimension (8 items), and resilience dimension (13 items). CD-RISC uses Likert 5 points scoring method, including "never" (0 points), "rarely" (1 point), "sometimes" (2 points), "often" (3 points), "always" (4 points), the total range of 0–100 points, the higher the score, the higher the level of psychological resilience. The total scores are used to classify psychological resilience into the following grades: excellent(> 80 points), good(71–80 points), average(60–70 points), poor(< 60 points). The Cronbach's alpha of the scale in our study was 0.943.

### 2.2.3 General Self-Efficacy Scale(GSES).
GSES was compiled by German clinical psychologist Schwarzer et al [27]. This study adopted the Chinese version translated by Wang et al. in 2001 [28]. Meanwhile, reliability, structural validity and predictive validity tests were conducted in the context of Chinese culture to verify the validity and reliability of the Chinese version of GSES. There are 10 items in the scale, and each item adopts the Likert 4 points scoring method, from "completely

incorrect" to "completely correct", the score is 1–4, and the total score is 10–40. The higher the score, the better the patient's self-efficacy. Self-efficacy is categorized into three levels based on measured scores: the low level is 10–19 points, the medium level is 20–30 points, and the high level is 31–40 points. The Cronbach's alpha of the scale in our study was 0.930.

**2.2.4 Self-Rating Anxiety Scale (SAS).** The SAS developed by Zung in 1971 was proved to be applicable to the study of anxiety in Chinese population through the measurement of reliability and validity [29]. Moreover, through exploratory factor and confirmatory factor analysis, the factors presented in different populations were stable. SAS consists of 20 questions, each of which is classified into four levels according to the frequency of symptoms. There were 15 negative statements (1–4 positive scoring) and 5 positive statements (4–1 reverse scoring), and the scores of each item are added together to get the total rough score (X). X was multiplied by the coefficient 1.25 and then rounded to get the standard score (Y). Based on SAS scores, anxiety levels are categorized as: normal (less than 50 points), mild (50–59 points), moderate (60–69 points), and severe (70 or greater points). The Cronbach's alpha of the scale in our study was 0.857.

**2.2.5 Self-Rating Depression Scale (SDS).** SDS was compiled by W.K.Zung in 1965, which can reflect the recent mood of the respondents to a certain extent. It was proved to be applicable to the study of depression in Chinese population through the exploratory factor and confirmatory factor analysis [30]. The scale includes 20 questions, and each question is divided into 4 levels according to the frequency of symptoms, including 10 positive score items and 10 reverse score items. The scoring rule is as follows: the score of the positive question was 1–4 points, and the score of the reverse question was 4–1 points. Sum the scores to get the total rough score (X), and multiply X by 1.25 to get the standard score (Y). The criteria for depression are as follows: SDS ≤ 52 points indicates no depression, 53–62 points indicates mild depression, 63–72 points indicates moderate depression, and 73–100 points indicates severe depression. The Cronbach's alpha of the scale in our study was 0.861.

## 2.3 Data collection

Electronic questionnaires were used to collect data. The researchers sent the questionnaire link to the participants, who scanned the QR code to fill out the online questionnaire.The questionnaire used unified guidance to explain the purpose of the survey, answer methods, filling requirements and precautions, etc. In order to ensure the quality of the survey results, all researchers had received uniform training and pre-survey training, and researchers work rigorously and communicate skillfully. Re-enter and check the data after collecting the questionnaire to ensure its authenticity and reliability.

## 2.4 Statistical analysis

Microsoft Excel 2019 was used for data entry, and IBM SPSS v25.0 was used for data analysis. The measurement data were described by means and standard deviations, while the count data were expressed as frequencies and percentages. Statistical inference was carried out according to the test level of $\alpha = 0.05$. Bilateral probability values were taken as $P$-values. In order to study the influence of demographic variables on psychological resilience, $T$-test, ANOVA and Kruskal-Wallis test were used. When comparing 3 or more independent groups, ANOVA were used for data with normal distribution and homogeneity of variance. Otherwise, Kruskal-Wallis test was used. To investigate the relationship among psychological resilience, self-efficacy, anxiety and depression, Pearson correlation coefficient test was adopted. To evaluate the strength of the correlation, we calculated the Pearson correlation coefficient ($r$) as the effect size. The value range of $r$ is from −1 to +1. The larger the absolute value, the stronger the correlation. Multiple linear regression analysis was used to predict the effects of demographic variables, self-efficacy, anxiety and depression on resilience. $P < 0.05$ was considered statistically significant.

## 2.5 Ethical considerations

This study was approved by the Ethics Committee of Xiangya Hospital, Central South University (registry number 2025010116). The study complied with the Declaration of Helsinki. All participants gave informed consent and their

participation was anonymous. Additionally, randomly generated numbers were used to anonymously tag participants and only authorized staff were allowed to view or process the data to improve data security and participant anonymity during online questionnaire administration. The online platform used for data collection complied with China's Cybersecurity Law.

## 3 Results

### 3.1 General information characteristics and univariate analysis on the influencing factors of psychological resilience

A total of 89 questionnaires were distributed while 89 questionnaires were valid, with an effective response rate of 100%. The average age of CMATMs was $41.37 \pm 7.10$, most of them were male (68.5%) and married(89.9%). The vast majority were medical staff (89.9%), and non-medical staff accounted for 10.1%. Postgraduate students accounted for 46.1% and the part-time staff accounted for 60.7%. Most of members had middle and senior titles, and the tertiary hospitals were mostly dispatched, accounting for 84.3%. 70.8% of the members received family support.

Univariate analysis showed that age, highest level of education, level of the dispatching organization, professional title, prior overseas experience before participating in foreign aid work, classification of hardship areas in recipient countries, and family support had statistical significance on psychological resilience scores. See Table 1 for details.

### 3.2 Participants' work and lifestyle in recipient countries and univariate analysis

The vast majority of CMATMs enjoyed satisfactory family life (95.5%) and work (93.3%). 74.2% of the members did not smoke and most of them exercise frequently (51.7%). More than half of them felt lonely (53.9%) and used the Internet for 1–4 hours per day (51.7%). The majority believed that the political situation (78.7%) and social security were basically stable (58.4%), the level of material conditions could basically meet their own needs (73.0%), they could adapt to the language environment (60.7%) and local climate (87.6%). Additionally, most people had been bitten by mosquitoes (82.0%) and infected with the COVID-19 (98.9%), while a small number had been infected with typhoid (4.5%) and malaria (9.0%). More than half of participants were directly involved in public health emergency relief missions in recipient countries (51.7%).

Univariate analysis showed that loneliness, cultural entertainment, frequency of exercise, economic subsidy policy, professional title promotion policy, social security situation in recipient countries, whether directly participated in public health emergency response missions in recipient countries would affect psychological resilience. See Table 2 for detailed information.

### 3.3 Psychological status of Chinese medical aid team members

Descriptive analysis showed that the total score of psychological resilience of CMATMs was generally at a good level. The three dimensions of optimism, self-strengthening and resilience were $25.37 \pm 4.39$, $10.18 \pm 2.63$, $36.98 \pm 7.69$, respectively. The total score of GSES was at the medium level. The total scores of SAS and SDS were at the normal level. The specific scores of each scale were shown in Table 3.

### 3.4 Correlation analysis of self-efficacy, anxiety, depression and psychological resilience

Pearson correlation analysis showed that self-efficacy was positively correlated with psychological resilience, while anxiety and depression were negatively correlated with psychological resilience. In addition, the higher the level of anxiety and depression, the lower the self-efficacy. Higher levels of anxiety were associated with increased symptoms of depression, and vice versa. See Table 4 for the interrelationships among psychological states.

### 3.5 The level of self efficacy, anxiety, and depression on psychological resilience

The results showed that the proportions of those with excellent, good, average and poor psychological resilience were 30.3%, 23.6%, 25.8, 20.2, respectively. Self-efficacy was mostly in the medium level(65.2%) and most people had no

**Table 1.** General information characteristics and the results of univariate analysis on the influencing factors of psychological resilience(n = 89).

| Variable | Category group | Frequency (%) | Mean (SD) | Univariate analysis results |
|---|---|---|---|---|
| **Age(years)** | ≤30 | 4(4.5) | 56.50(12.48) | $F = 3.643$ $P = 0.016^*$ |
| | 31-40 | 43(48.3) | 71.81(14.11) | |
| | 41-50 | 31(34.8) | 72.61(12.22) | |
| | >50 | 11(12.4) | 80.91(9.26) | |
| **Sex** | Male | 61(68.5) | 72.93(14.03) | $t = 0.418$ $P = 0.677$ |
| | Female | 28(31.5) | 71.64(12.40) | |
| **Marital status** | Spinsterhood | 7(7.9) | 65.29(18.71) | $F = 1.209$ $P = 0.304$ |
| | Married | 80 (89.9) | 73.04(12.82) | |
| | Others | 2(2.2) | 77.5(21.92) | |
| **Highest level of education** | Bachelor degree | 27(30.3) | 60.11(9.77) | $F = 28.818$ $P = 0.000^*$ |
| | Master degree | 41(46.1) | 76.02(10.57) | |
| | Doctor's degree or above | 21(23.6) | 81.67(11.46) | |
| **Occupation** | Medical staff | 80(89.9) | 73.41(13.46) | $F = 0.473$ $P = 0.065$ |
| | Non-medical personnel | 9(10.1) | 64.67(11.58) | |
| **Professional title** | Junior | 3(3.4) | 49.00(0.00) | $H = 27.151$ $P = 0.000^*$ |
| | Intermediate | 27(30.3) | 64.07(11.59) | |
| | Senior | 59(66.3) | 77.59(11.29) | |
| **Level of the dispatching organization** | Non- tertiary hospitals | 14(15.7) | 57.79(11.26) | $t = -5.037$ $P = 0.000^*$ |
| | Tertiary hospitals | 75(84.3) | 75.28(12.04) | |
| **Prior overseas experience before participating in foreign aid work** | No | 47(52.8) | 68.00(12.56) | $t = -3.568$ $P = 0.001^*$ |
| | Yes | 42(47.2) | 77.60(12.78) | |
| **Family fertility** | Childlessness | 12(13.5) | 72.00(17.88) | $F = 0.020$ $P = 0.980$ |
| | One child | 42(47.2) | 72.81(12.98) | |
| | Second children and above | 35(39.3) | 72.37(12.79) | |
| **Family support** | Oppose | 13(14.6) | 56.85(6.84) | $H = 25.424$ $P = 0.000^*$ |
| | Neutrality | 13(14.6) | 68.23(15.20) | |
| | Support | 63(70.8) | 76.65(11.47) | |
| **Classification of hardship areas in recipient countries^** | ClassI | 9(10.1) | 51.89(3.76) | $H = 38.011$ $P = 0.000^*$ |
| | ClassII | 19(21.3) | 67.89(10.52) | |
| | ClassIII | 24(27.0) | 72.00(10.62) | |
| | ClassIV | 30(33.7) | 77.83(11.37) | |
| | ClassV | 7(7.9) | 90.71(4.79) | |

Note:

*indicates $P < 0.05$, SD stands for standard deviation,

^stands for the classification standards of hardship areas in recipient countries formulated by the Ministry of Finance of China, and the higher the grade, the higher the hardship.

anxiety (85.4%) or depression (83.1%). The Kruskal-Wallis $H$ test showed that self-efficacy, anxiety and depression of different degrees had significant effects on psychological resilience ($p < 0.05$). See Table 5 for details.

### 3.6 Multivariate analysis of psychological resilience

To identify factors associated with psychological resilience, multiple linear regression analysis was conducted. The total score on the psychological resilience scale served as the dependent variable. The demographic variables with statistical significance in the univariate analysis and the total scores on the self-efficacy, anxiety, and depression scales were taken

**Table 2. Work and lifestyle in recipient countries and and the results of univariate analysis on the influencing factors of psychological resilience(n = 89).**

| Variable | Category group | Frequency (%) | Mean (SD) | Univariate analysis results |
|---|---|---|---|---|
| Smoking status | No | 66(74.2) | 71.56(13.55) | $t=-1.149$ $P=0.254$ |
| | Yes | 23(25.8) | 75.30(13.17) | |
| Frequency of drinking | Everyday | 2(2.2) | 56.00(9.90) | $F=1.243$ $P=0.299$ |
| | Per week | 24(27.0) | 72.21(12.65) | |
| | Monthly | 26(29.2) | 74.65(15.50) | |
| | No drinking | 37(41.6) | 72.14(12.41) | |
| Frequency of exercise | Never | 5(5.6) | 58.80(9.50) | $F=4.102$ $P=0.009^*$ |
| | Now and then | 18(20.2) | 67.33(15.37) | |
| | Frequently | 46(51.7) | 73.85(12.12) | |
| | Almost every day | 20(22.5) | 77.60(12.54) | |
| Average daily Internet use time | <1 hour | 5(5.6) | 73.40(15.63) | $F=0.085$ $P=0.919$ |
| | 1-4 hours | 46(51.7) | 71.96(13.60) | |
| | >4 hours | 38(42.7) | 73.10(13.42) | |
| Loneliness | Yes | 48(53.9) | 64.44(9.88) | $t=-8.042$ $P=0.000^*$ |
| | No | 41(46.1) | 82.00(10.71) | |
| Culture and entertainment | Dissatisfy | 8(9.0) | 62.63(11.75) | $F=9.153$ $P=0.000^*$ |
| | Neutral | 56(62.9) | 70.23(12.98) | |
| | Satisfaction | 25(28.1) | 80.84(11.10) | |
| Economic subsidy policy | Dissatisfy | 3(3.4) | 50.33(2.31) | $F=5.484$ $P=0.006^*$ |
| | Neutral | 56(62.9) | 71.98(13.18) | |
| | Satisfaction | 30(33.7) | 75.77(12.63) | |
| Professional title promotion policy | Dissatisfy | 6(6.7) | 60.33(8.82) | $F=9.174$ $P=0.000^*$ |
| | Neutral | 42(47.2) | 68.74(12.19) | |
| | Satisfaction | 41(46.1) | 78.20(12.95) | |
| Social security situation in recipient countries | Very unstable | 5(5.6) | 86.00(9.19) | $F=4.306$ $P=0.007^*$ |
| | Instability | 28(31.5) | 67.89(10.94) | |
| | Basically stable | 52(58.4) | 72.79(14.00) | |
| | Very stable | 4(4.5) | 84.75(10.21) | |
| Direct participation in public health emergency response missions in recipient countries | No | 43(48.3) | 79.51(12.73) | $t=5.438$ $P=0.000^*$ |
| | Yes | 46(51.7) | 66.00(10.68) | |

Note:

*indicates $P<0.05$, SD stands for standard deviation.

as independent variables. After the residual test, the histogram and normal P-P plot for regression of standardized residuals were roughly normally distributed. The multicollinearity test found that VIF values were all less than 10, indicating that the possibility of multicollinearity between variables was small. The adjusted R² for the multivariate model was 0.850, indicating that the model explained approximately 85.0% of the variation degree of psychological resilience, and the model had a relatively high goodness of fit.

Results showed that self-efficacy, professional title, loneliness, classification of hardship areas in recipient countries, family support and cultural entertainment were positively correlated with psychological resilience, while direct participation in public health emergency response missions in recipient countries was negatively correlated with psychological resilience. See Table 6 for specific information and assignment.

**Table 3. Psychological status scores of Chinese medical aid team members.**

| Scale | Total score (Mean±SD) |
|---|---|
| CD-RISC | 72.53±13.48 |
| GSES | 28.73±5.36 |
| SAS | 40.01±9.56 |
| SDS | 41.64±10.07 |

Note: SD stands for standard deviation, CD-RISC stands for The Connor Davidson Resilience Scale, GSES stands for General Self-Efficacy Scale, SAS stands for Self-Rating Anxiety Scale, SDS stands for Self-Rating Depression Scale.

**Table 4. Correlation analysis of self-efficacy, anxiety, depression and psychological resilience (r value).**

| Variable | Psychological resilience | Self-efficacy | Anxiety | Depression |
|---|---|---|---|---|
| Psychological resilience | 1 | 0.754** | −0.298** | −0.470** |
| Self-efficacy | 0.754** | 1 | −.093 | −0.285** |
| Anxiety | −0.298** | −0.093 | 1 | 0.802** |
| Depression | −0.470** | −0.285** | 0.802** | 1 |

Note:

**indicates $P < 0.01$.

**Table 5. The level of self efficacy, anxiety, and depression on psychological resilience.**

| Variable | Category group | Frequency (%) | Kruskal-Wallis H |
|---|---|---|---|
| Psychological resilience | Excellent | 27(30.3) | —— |
| | Good | 21(23.6) | |
| | Average | 23(25.8) | |
| | Poor | 18(20.2) | |
| Self-efficacy | Low level | 4(4.5) | $H = 43.398$ $P = 0.000^*$ |
| | Medium level | 58(65.2) | |
| | High lever | 27(30.3) | |
| Anxiety | Normal | 76 (85.4) | $H = 9.791$ $P = 0.007^*$ |
| | Mild | 8(9.0) | |
| | Moderate | 5(5.6) | |
| | Severe | 0 | |
| Depression | Normal | 74(83.1) | $H = 10.977$ $P = 0.004^*$ |
| | Mild | 9(10.1) | |
| | Moderate | 6(6.7) | |
| | Severe | 0 | |

Note:

*indicates $P < 0.05$

## 4 Discussion

As far as we know, this study is the first to explore the influencing factors of psychological resilience of CMATMs under public health emergencies. In this study, we identified a positive and significant relationship between psychological resilience and self-efficacy, but a negative correlation with anxiety and depression. In addition, other factors that influence psychological resilience were identified, including professional title, loneliness, classification of hardship areas in recipient

**Table 6. Multivariate analysis of psychological resilience.**

| Variable | B-Value | Standard Error | t | P | VIF | Assignment |
|---|---|---|---|---|---|---|
| Self-efficacy | 1.024 | 0.133 | 7.675 | 0.000 | 1.655 | —— |
| Professional title | 4.679 | 1.245 | 3.758 | 0.001 | 1.524 | "Junior"=1,"Intermediate"=2, "Senior"=3 |
| Loneliness | 4.973 | 1.453 | 3.422 | 0.000 | 1.715 | "Yes"=1, "No"=2 |
| Classification of hardship areas in recipient countries^ | 1.864 | 0.632 | 2.951 | 0.004 | 1.649 | "ClassI"=1, "ClassII"=2, "ClassIII"=3, "Class IV "=4, "Class V"=5 |
| Family support | 3.110 | 1.059 | 3.303 | 0.001 | 1.561 | "Oppose"=1, "Neutrality"=2, "Support"=3 |
| Cultural and entertainment | 3.537 | 0.942 | 3.340 | 0.001 | 1.226 | "Dissatisfy"=1, "Neutral"=2; "Satisfaction"=3 |
| Direct participation in public health emergency response missions in recipient countries | −3.775 | 1.325 | −2.849 | 0.006 | 1.434 | "No"=1 "Yes"=2 |

Note: VIF indicates Variance Inflation Factor,

^stands for the classification standards of hardship areas in recipient countries formulated by the Ministry of Finance of China, and the higher the grade, the higher the hardship.

countries, family support, cultural and entertainment, direct participation in public health emergency response missions in recipient countries.

Our study have found that the higher the self-efficacy of CMATMs, the better the psychological resilience. This is similar to the study of Berdida et al. [21], who suggest that nurses with a strong sense of self-efficacy show a lower level of emotional exhaustion, in turn, improving self-efficacy can improve the level of resilience. In addition, previous studies have shown that medical staff with poor psychological resilience are more likely to suffer from depression and anxiety [31,32], which is consistent with our findings. Under public health emergency, team members may feel vulnerable and out of control, which may reduce their sense of self-efficacy and induce anxiety and depression. Relevant studies have shown that mindfulness based strategies have measurable results in reducing burnout and enhancing the resilience of healthcare professionals [16,33]. Therefore, we can use mindfulness based strategies to implement relevant psychological interventions. Specifically, we can introduce the concept, principles, and importance of mindfulness to facilitate understanding of its core elements. These concepts were then reinforced through practical exercises such as meditation, gentle yoga, music, and breathing, enabling participants to experience a state of mindfulness and integrate it into their daily routines. It is also possible to offer systematic courses, set up mindfulness groups and strengthen communication among team members. The implementation of these interventions can reduce the fatigue of team members, improve work participation and self-regulation.

An interesting finding of our study was that CMATMs who were directly involved in emergency public health rescue missions showed worse psychological resilience. Public emergencies are inherently characterized by high-stress environments compounded by a multitude of trauma scenes, including mass casualties, severe illness, and the suffering and death of patients [34]. Team members directly involved in the response face life-and-death decisions, patient suffering and threats to their own safety for long periods of time, which can trigger acute stress reactions or post-traumatic stress disorder (PTSD). Repeated exposure to other people's traumatic experiences can also lead to vicarious trauma, causing emotional numbness, anxiety or depression, and weakening psychological resilience. Unlike routine medical practice, these situations often involve unpredictable circumstances, such as resource scarcity, ethical dilemmas surrounding patient prioritization, and the potential for exposure to highly contagious pathogens. Rescue workers need to make a large number of decisions in a short period of time, such as determining the priority order for treating the injured and choosing the best rescue plan, etc. Long-term and high-intensity decision-making may lead to the depletion of cognitive resources, reduce an individual's self-control ability and the ability to cope with stress, thereby affecting psychological resilience. Relevant

studies have shown that it is very difficult to participate in rescue tasks in emergency situations [35], which has a negative impact on the psychology of team members. According to the United Nations Office for the Coordination of Humanitarian Affairs and the Aid Worker Security Database, at least 281 aid workers have been killed worldwide in 2024 [36]. Maleki et al. believed that the perception of job insecurity would make nurses prone to physical fatigue and psychological problems [37]. With the increase of risk perception, adaptive behaviors at the individual and social levels will increase [38]. Therefore, mental health service institutions should be established locally. Verified psychological resilience assessment tools should be adopted to regularly screen the psychological resilience levels of team members [12]. This allows for the early identification of those with low resilience or potential psychological risks, facilitating the delivery of timely and targeted interventions to improve their coping skills in the face of trauma and stress. In addition, institutions should establish personal psychological files for team members, recording the screening results, intervention measures and effects, providing a basis for subsequent mental health management.

Our research also found that members with higher professional titles had better psychological resilience. Young people are usually more prone to stress, possibly because of their lack of relevant work and social experience. Furthermore, perceived inequities in welfare benefits compared to senior colleagues can result in feelings that their work is undervalued and their career prospects are limited. Team members with higher professional titles usually have greater job autonomy and a sense of perceived control. They can arrange their work content and decide on work methods more independently, believing that they can control the work environment and cope with various challenges. Additionally, members with higher professional titles may be more likely to receive emotional support and protection from leaders, which can help them better cope with the challenges in medical aid work. Therefore, dispatching units should encourage them to participate in the formulation of medical plans, and allow them to adjust the treatment plans according to the local actual situation, so as to enhance the autonomy of the work. In addition, the training for the leaders of team members should be strengthened to enhance their leadership and management capabilities. Leaders should provide emotional support and psychological counseling for the team members, and increase their sense of perception and control. At the same time, it is necessary to create a supportive working environment for young or inexperienced CMATMs, such as adding the content of mental health knowledge in the onboarding training, improving promotion policies to promote their career development, reasonably and flexibly arranging working hours.

The research showed that cultural entertainment conditions of recipient countries are positively correlated with psychological resilience. Chen et al. [8] has found that the entertainment situation of CMATMs is not so positive, which may be due to limited social circles, monotonous activities and a lack of leisure time caused by high-intensity work. Our study also found that team members in difficult recipient country and those with pronounced sense of loneliness had worse psychological resilience, which is consistent with the study of Chen et al. [7]. Previous studies have shown that foreign aid workers tend to go to countries with high economic levels [39]. However, the level of medical technology in recipient countries is usually worse than that of China. It is difficult to obtain advanced and up-to-date equipment and technology during the period of foreign aid. The difficult living conditions have a negative impact on their willingness to go abroad and their coping style. Bjerneld et al. [40] conducted interviews with Swedish medical personnel involved in health assistance and found that harsh conditions, arduous tasks and language barriers were stressors for respondents, who felt isolated by external and local social isolation. To promote successful cross-cultural adjustment, team leaders and local institutions should regularly organize psychological and social situation introduction meetings based on cultural differences in different countries and regions. More attention should be paid to the entertainment life of CMATMs, their work burden and sense of loneliness can be alleviated through the provision of knowledge and psychological training, as well as a variety of team-building activities. Additionally, our study confirms that family support is an important protective factor for good psychological resilience of team members, which is consistent with the study of Rothon et al. [41]. Work-family conflict is the main reason hindering international assignment, and lack of family support is positively correlated with depressive symptoms of employees [7,42]. Therefore, superior leaders should mobilize the family and social support network to strengthen

the comprehensive support for CMATMs [43], covering welfare benefits, professional title evaluation, infrastructure, job promotion mental health, family care and other aspects appropriate preferential treatment. Relevant institutions should establish smooth information channels, organize regular family symposiums, set up family support hotlines, provide psychological counseling and support services for team members and their families, eliminating their doubts and concerns. At the same time, the dispatch and rotation mechanism of foreign aid medical team members should be improved, and personalized dispatch plans can be provided for a period of time. For example, the family visit leave for team members can be appropriately increased, or they can be allowed to return to their home country for leave in advance after completing a certain amount of work.

## 5 Limitations

While the study provides important evidence for improving psychological resilience in CMATMs, it also found some limitations. First, the sample size is relatively small and the subjects are all Chinese, which may limit the extrapolation of the findings. However, due to the particularity of the foreign medical aid team members, we could not obtain enough sample size, which makes us cautious in interpreting the research results. Second, our scales are self-report scales, which can lead to potential response bias, such as social desirability bias in self-reported measures. In the future, peer-report scales should be further developed to evaluate the psychological state of CMATMs. Finally, team members are sent to multiple countries with different cultural backgrounds and infrastructure that cannot be explored in a single article.

## 6 Conclusions

This study revealed that CMATMs exhibited lower psychological resilience when characterized by a low sense of self-efficacy, a lower professional title, direct involvement in emergency public health rescue efforts, pronounced sense of loneliness, and limited access to cultural and recreational activities in the recipient area. CMATMs are key professionals in China's foreign aid, this study provides guidance for the relevant institutions to select appropriate intervention measures according to the influence factors, which can help team members enhance their ability to cope with stress, maintain a good working state and efficiency, and thereby enhance the cohesion and collaboration ability of the foreign aid team. Policymakers or healthcare administrators can design resilience assessment programs and workshops, strengthen mental health training, mobilize family and social support networks, improve infrastructure and welfare security policies. Multi-country comparative studies with large sample using longitudinal methods are needed to investigate the detailed factors affecting psychological resilience and explore the sociocultural adaptation of CMATMs in different countries in the future.

## Supporting information

**S1 File. The raw data.**
(XLSX)

## Author contributions

**Conceptualization:** Xueyi Wei, Yinhua Liang, Jun Huang.

**Data curation:** Xueyi Wei, Yinhua Liang, Jun Huang.

**Formal analysis:** Xueyi Wei, Yinhua Liang.

**Methodology:** Xueyi Wei, Yinhua Liang.

**Software:** Xueyi Wei.

**Writing – original draft:** Xueyi Wei.

**Writing – review & editing:** Xueyi Wei, Yinhua Liang, Jun Huang.

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
