## [Decision Letter · Decision Letter 0]

Dear Dr. xueyi,

Thank you for submitting your manuscript to PLOS ONE. After careful consideration, we feel that it has merit but does not fully meet PLOS ONE’s publication criteria as it currently stands. Therefore, we invite you to submit a revised version of the manuscript that addresses the points raised during the review process.

Please submit your revised manuscript by Apr 17 2025 11:59PM to ensure timely processing and further consideration for publication. If you will need more time than this to complete your revisions, please reply to this message or contact the journal office at plosone@plos.org . A rebuttal letter that responds to each point raised by the academic editor and reviewer(s). You should upload this letter as a separate file labeled 'Response to Reviewers'.A marked-up copy of your manuscript that highlights changes made to the original version. You should upload this as a separate file labeled 'Revised Manuscript with Track Changes'.An unmarked version of your revised paper without tracked changes. You should upload this as a separate file labeled 'Manuscript'.

We look forward to receiving your revised manuscript.

Kind regards,

Prof. Ebtsam Aly Abou Hashish,

Academic Editor

PLOS ONE

 [This work was funded by the National Natural Science Foundation of China [82160554], Natural Science Foundation of Hunan Province [2020JJ8051]. And supported by the Project Program of National Clinical Research Center for Geriatric Disorders [Xiangya Hospital, Grant No. 2021LNJJ15].]. 

Additional Editor Comments:

Dear Authors,

Thank you for your submission to PLOS ONE. The reviewers have provided constructive feedback to enhance the clarity, rigor, and impact of your manuscript. While your study presents valuable insights into preoperative readiness among patients undergoing arteriovenous fistula percutaneous transluminal angioplasty, several areas require revision to strengthen the manuscript's coherence, methodological transparency, and discussion depth.

Reviewer 1 highlights the need for aligning the manuscript with journal guidelines, particularly in reporting methodological details, defining the research problem, and emphasizing the study’s practical applications. Additionally, more comparisons with previous studies in the discussion section, along with citations from suggested references, will improve the scholarly context of your work.

Reviewer 2 suggests refining the abstract by specifying the study duration and clarifying its implications. The introduction would benefit from a smoother transition into the psychological aspects of preoperative readiness, and more recent references should be included. The methodology section requires additional details on participant selection, sample size justification, scale reliability, and the rationale for statistical choices. In the results, key findings should be highlighted more explicitly, and assumptions for multivariate analysis should be addressed. The discussion needs a deeper exploration of the correlation between psychological resilience and preoperative readiness, along with more actionable recommendations for practice. Finally, the conclusion should present specific strategies for healthcare implementation, and minor grammatical improvements should be made throughout.

To proceed, please revise your manuscript accordingly, ensuring that:

The abstract clearly specifies the study period and includes practical implications.

The introduction provides a well-defined research problem, the significance of the study variables, and a clear rationale for conducting the study.

The methods section includes details on sampling methods, scale reliability, validation procedures, and justification for statistical analyses.

The results section presents findings clearly with sufficient interpretation and proper statistical reporting, including effect sizes.

The discussion integrates more comparative studies, explores psychological resilience more deeply, and includes targeted intervention strategies.

The conclusion offers specific recommendations for future research and policy implications.

The language and formatting are refined to improve clarity and precision.

We encourage you to carefully address the reviewers' concerns and incorporate their suggested references to strengthen your study’s academic contribution. Once revised, please submit a detailed response letter explaining how each point was addressed.

We look forward to receiving your revised manuscript.

Best regards,

Ebtsam Abou Hashish

Reviewers' comments:

Reviewer's Responses to Questions

**Comments to the Author**

1. Is the manuscript technically sound, and do the data support the conclusions?

Reviewer #1: Yes

Reviewer #2: Yes

2. Has the statistical analysis been performed appropriately and rigorously?

Reviewer #1: Yes

Reviewer #2: Yes

3. Have the authors made all data underlying the findings in their manuscript fully available?

Reviewer #1: Yes

Reviewer #2: Yes

4. Is the manuscript presented in an intelligible fashion and written in standard English?

Reviewer #1: Yes

Reviewer #2: Yes

Reviewer #1: This is an interesting paper and the content is very good but I suggest the paper can be improved in the following ways: Abstract

-Please correct all parts of the article according to the guidelines of the journal authors guideline In the methods section please bring year of performing of this study, sampling methods and data analysis methods -In the conclusion part, it is necessary to specify the researcher's proposal to improve the conditions and use of the beneficiaries

Introduction

Please bring the following items 1- Definition of the research problem 2- The magnitude and importance of the study variable 3- Expressing the necessity of conducting the study Finally, the practical purpose of the study should be stated. Methods

Please report the details scoring and validations of study tools

Discussion

In the discussion section, it is necessary y to compare the main results of the study with the results of other studies in this field. To strengthen the article, especially in the introduction and discussion section the following studies are suggested, please used and add to this manuscript references.

-Crowd Simulations and Determining the Critical Density Point of Emergency Situations

-Heat waves and adaptation: A global systematic review

_Medical tourism development: A systematic review of economic aspects

- Investigating the burden of disease dimensions (time-dependent, developmental, physical, social and emotional) among family caregivers with COVID-19 patients

Conclusion � What are your suggestion for future studies? Best regards

Reviewer #2: Dear authors

Thank you for your efforts to do this research. Below are some comments and suggestions to help improve the clarity and impact of your study.

Abstract:

Please specify the study duration.

The abstract provides a concise summary, but it lacks clarity in presenting the study's implications. Including a brief statement on how the findings can inform interventions would strengthen its impact.

The conclusion section in the abstract could be more actionable, specifying the next steps or direct applications of the findings.

Introduction section:

The transition to psychological resilience as a focus needs smoother integration. Consider elaborating on why psychological resilience, specifically, was prioritized over other psychological constructs.

References are well-cited but could benefit from the inclusion of more recent or diverse studies for broader relevance. For example, the authors can refer to the following articles on this topic:

https://www.mdpi.com/1660-4601/18/23/12548

https://www.frontiersin.org/journals/public-health/articles/10.3389/fpubh.2022.1034624/full

Methods:

The inclusion criteria are clear, but more details about the rationale for participant selection would be beneficial. Why was the range of roles (e.g., cooks and translators) included, and how do these roles contribute to the overall findings?

The sample size (n=89) is relatively small. The limitations section addresses this, but a justification earlier in the methods could preempt concerns.

Please explain the sampling method and how the sample size was estimated.

The use of validated scales like CD-RISC, GSES, SAS, and SDS is commendable. However:

Provide reliability coefficients (e.g., Cronbach's alpha) for these scales in the study sample, not just general references. Clarify if the scales were culturally adapted for Chinese participants beyond translation (e.g., if factor structures were re-validated).

Statistical methods are appropriate but require clarification: Explain why certain tests (e.g., ANOVA vs. Kruskal-Wallis) were chosen for specific analyses.

Include effect sizes alongside p-values to provide a more robust interpretation of significant results.

Results:

Ensure all abbreviations used in tables are defined in the captions or footnotes.

Highlight key findings directly in the text rather than leaving them for the reader to infer from the tables.

The reported correlations between psychological resilience and variables like self-efficacy, anxiety, and depression are meaningful. However, the strength of these relationships should be contextualized within existing literature.

Multivariate Analysis: While the regression model is insightful, assumptions underlying the regression (e.g., multicollinearity, normality of residuals) are not addressed. Adding these details would strengthen the validity of the findings.

Discussion Section:

The negative correlation between direct participation in public health emergencies and psychological resilience deserves deeper exploration. Was this due to trauma exposure, workload, or other factors?

Suggestions for interventions, such as mindfulness strategies, are useful but could be made more specific and actionable.

The limitations are honestly presented but could be expanded to discuss potential response biases (e.g., social desirability bias in self-reported measures).

I suggest enriching the discussion by addressing related studies. For example, I recommend the following article:

https://journals.lww.com/jnmr/fulltext/2024/29030/relationship_between_resilience_and_social_and.12.aspx?context=latestarticles

Conclusion:

The conclusion reiterates key findings but lacks actionable recommendations for policymakers or healthcare administrators. Consider framing specific strategies based on identified factors (e.g., training modules, resilience workshops).

References:

The references are current and relevant, but ensure consistency in formatting (e.g., capitalization, use of journal abbreviations).

Ethical Considerations:

Ethical approval and participant consent are adequately described. Consider briefly addressing how data security and participant anonymity were ensured during online questionnaire administration.

Language and Grammar:

The manuscript is generally well-written but has minor grammatical issues. For instance, "strong sense of loneliness" could be revised to "pronounced sense of loneliness" for precision.

In line 83, replace "are" with "were."

**Do you want your identity to be public for this peer review?** For information about this choice, including consent withdrawal, please see our Privacy Policy

Reviewer #1: No

Reviewer #2: No

---

## [Author Response · Author response to Decision Letter 1]

9 Mar 2025

Reviewer #1: This is an interesting paper and the content is very good but I suggest the paper can be improved in the following ways: Abstract

-Please correct all parts of the article according to the guidelines of the journal authors guideline In the methods section please bring year of performing of this study, sampling methods and data analysis methods -In the conclusion part, it is necessary to specify the researcher's proposal to improve the conditions and use of the beneficiaries

Answer: Thank you for your valuable feedback.

We have revised all parts of the article according to the guidelines of the journal authors guideline. Additionally, we have supplemented the year of performing of this study, sampling methods and data analysis methods of this study in the section of methods. In the conclusion part, the specific suggestions made by the researchers are also added.

Introduction

Please bring the following items 1- Definition of the research problem 2- The magnitude and importance of the study variable 3- Expressing the necessity of conducting the study Finally, the practical purpose of the study should be stated.

Answer: Thank you for your valuable insights regarding our manuscript.

1-According to your suggestion, we have reorganized the logic of the introduction and clarified our research question. There are two main research questions, which are the status quo of psychology resilience of foreign aid medical team members and the factors affecting their psychology resilience.

2-We added the magnitude and importance of study variables as you suggested, such as psychology resilience, self-efficacy, social support, etc.

3- We have explained the practical purpose of the study at the end.

This study aims to reveal the level of psychological resilience of this population and its related factors, with the goal of providing theoretical basis for effectively improving the psychological status of CMATMs and reducing the risk of mental diseases. Through the comprehensive analysis of these influencing factors, not only can provide a more accurate assessment of the psychological resilience of CMATMs, but also provide a scientific basis for follow-up intervention and support. Finally, it will help improve the overall adaptability and work stability of CMATMs, ensure the smooth development of foreign medical aid work, promote international medical cooperation and exchanges, enhance the friendship and mutual trust between different countries.

Methods

Please report the details scoring and validations of study tools

Answer: Thank you for your valuable feedback.

We have reported on the details of the study tools in section 2.2.2-2.2.5, such as scoring criteria and Cronbach's alpha coefficient.

Discussion

In the discussion section, it is necessary to compare the main results of the study with the results of other studies in this field. To strengthen the article, especially in the introduction and discussion section the following studies are suggested, please used and add to this manuscript references.

-Crowd Simulations and Determining the Critical Density Point of Emergency Situations

-Heat waves and adaptation: A global systematic review

_Medical tourism development: A systematic review of economic aspects

- Investigating the burden of disease dimensions (time-dependent, developmental, physical, social and emotional) among family caregivers with COVID-19 patients

Answer: Thank you for your careful feedback.

We have compared the main results of the study with the results of other studies in this field. Additionally, We have added a large number of references in accordance with your comments, respectively in reference 11, reference 13,reference 31, and reference 34.

Conclusion What are your suggestion for future studies? Best regards

Answer: Thank you for raising this point.

We have added our suggestion for future studies in the conclusion section.

Multi-center study with large sample studies are needed to investigate the detailed factors affecting psychological resilience and explore the sociocultural adaptation of CMATMs in different countries in the future.

Reviewer #2: Dear authors

Thank you for your efforts to do this research. Below are some comments and suggestions to help improve the clarity and impact of your study.

Abstract:

Please specify the study duration.

Answer: Thank you for your suggestion.

We have added the specific study duration in the methods. This study was conducted from 3rd December, 2024 to 25th December, 2024.

The abstract provides a concise summary, but it lacks clarity in presenting the study's implications. Including a brief statement on how the findings can inform interventions would strengthen its impact.

Answer: Thank you for your valuable feedback.

We added a brief statement on how the findings can inform interventions would strengthen its impact. This study provides a theoretical basis for formulating psychological intervention measures according to the factors that affect the psychological resilience of CMATMs to help them establish a more adaptive defense mechanism and maintain good psychological resilience during foreign aid.

The conclusion section in the abstract could be more actionable, specifying the next steps or direct applications of the findings.

Answer: Thank you for your thoughtful comments.

We have revised our conclusion section to specify the next steps or direct applications of the findings.

Policymakers or healthcare administrators should regular assess CMATMs ' mental health status, strengthen mental health training, mobilize family and social support networks, improve infrastructure and welfare security policies.

Introduction section:

The transition to psychological resilience as a focus needs smoother integration. Consider elaborating on why psychological resilience, specifically, was prioritized over other psychological constructs.

Answer: Thank you for your insightful feedback.

We have revised the introduction to better integrate the focus on psychological resilience. In this revision, we emphasize the importance of resilience in the context of the challenges faced by foreign medical aid teams. These teams often encounter a wide range of psychological issues, and one of the key strategies for effectively coping with stress and maintaining mental health is resilience. This adjustment aims to provide a smoother transition and a clearer rationale for prioritizing psychological resilience in our study. We appreciate your suggestion, which has helped to improve the manuscript's coherence and focus.

References are well-cited but could benefit from the inclusion of more recent or diverse studies for broader relevance. For example, the authors can refer to the following articles on this topic:

https://www.mdpi.com/1660-4601/18/23/12548

https://www.frontiersin.org/journals/public-health/articles/10.3389/fpubh.2022.1034624/full

Answer: Thank you for your valuable suggestion.

We have carefully read and analyzed the references you have provided. We also have already included recommended references in the revised manuscript and cited them as reference 12 and reference 33 to enhancing the quality of our manuscript.

Methods:

The inclusion criteria are clear, but more details about the rationale for participant selection would be beneficial. Why was the range of roles (e.g., cooks and translators) included, and how do these roles contribute to the overall findings?

Answer: Thank you for your valuable feedback.

We have added the reasons for selecting non-medical members to the inclusion criteria, and in order to harmonize terminology, we will include medical and non-medical members in the study. While medical professionals were the core of CMATMs, the non-medical members were often an important support force of the team, which had an indirect impact on the emotional and mental health of the medical professionals. In order to fully understand the psychological resilience of the entire team, our study was not limited to the core medical staff, but also included non-medical staff.

The sample size (n=89) is relatively small. The limitations section addresses this, but a justification earlier in the methods could preempt concerns.

Answer: Thank you for your valuable insights regarding our manuscript.

We have added a justification for the sample size in Section 2.1 Study Design and participants, where we explain the sample size calculation method and provide reasons for the relatively small sample size. Due to the special research population of CMATMs, the number was relatively small. In addition, CMATMs were distributed in different countries and regions, it was difficult for researchers to contact all potential participants. Therefore, we did our best to select only 89 subjects. Relatively small sample size was undeniably a limitation of our research. This additional explanation aims to preempt concerns and clarify the rationale behind the chosen sample size.

Please explain the sampling method and how the sample size was estimated.

Answer: Thank you for your valuable feedback.

We have added the sampling method and the sample size calculation method in Section 2.1 Study Design and participants.

The use of validated scales like CD-RISC, GSES, SAS, and SDS is commendable. However:

Provide reliability coefficients (e.g., Cronbach's alpha) for these scales in the study sample, not just general references. Clarify if the scales were culturally adapted for Chinese participants beyond translation (e.g., if factor structures were re-validated).

Answer: Thank you for your attention to these details.

We have added the reliability coefficients in section 2.2.2-2.2.5, including Cronbach's alpha, for the scales used in our study. These coefficients are based on the data from our study sample, not just general references. Additionally, we have clarified that the scales were culturally adapted and re-validated for use with Chinese participants to enhance the transparency and rigor of our study.

Statistical methods are appropriate but require clarification: Explain why certain tests (e.g., ANOVA vs. Kruskal-Wallis) were chosen for specific analyses.

Include effect sizes alongside p-values to provide a more robust interpretation of significant results.

Answer: Thank you for your thoughtful suggestion.

We have added further clarification in Section 2.4 regarding the rationale for selecting the statistical methods used in our study. Additionally, we have included effect sizes alongside p-values to offer a more comprehensive interpretation of the significant results. These revisions aim to improve the clarity and robustness of the statistical analysis.

In order to study the influence of demographic variables on psychological resilience, T-test ANOVA and Kruskal-Wallis test were used. Depending on the normality of the data, when comparing 3 or more independent groups, ANOVA were used for data with normal distribution and homogeneity of variance. Otherwise, Kruskal-Wallis test was used. To investigate the relationship among psychological resilience, self-efficacy, anxiety and depression, Pearson correlation coefficient test was adopted and to predict the effects of demographic variables, self-efficacy, anxiety and depression on resilience, multiple linear regression analysis was used. The significance level for all variables was considered to be 0.05.

Results:

Ensure all abbreviations used in tables are defined in the captions or footnotes.

Answer: Thank you for your reminder.

We have revised all abbreviations used in the tables to enhance the clarity and accessibility of the manuscript for readers.

Highlight key findings directly in the text rather than leaving them for the reader to infer from the tables.

Answer: Thank you for your suggestion.

We have revised the manuscript to highlight the key findings directly in the text to improve the readability and ensure that the main outcomes are clearly emphasized for the reader.

The reported correlations between psychological resilience and variables like self-efficacy, anxiety, and depression are meaningful. However, the strength of these relationships should be contextualized within existing literature.

Answer: Thank you for your valuable feedback.

We have expanded the discussion section to provide context for the reported correlations between psychological resilience and variables like self-efficacy, anxiety, and depression. Specifically, we have referenced existing literature to help contextualize the strength of these relationships and provide a deeper understanding of their significance.

Multivariate Analysis: While the regression model is insightful, assumptions underlying the regression (e.g., multicollinearity, normality of residuals) are not addressed. Adding these details would strengthen the validity of the findings.

Answer: Thank you for your valuable feedback.

We have assessed multicollinearity using Variance Inflation Factors (VIF) and ensured that no significant issues were found. The test results are supplemented in Section 3.6 and Figure 6. Additionally, we have tested the normality of residuals using the residual test and visualized it with histogram and P-P plot. These checks confirm that the assumptions for the regression model are adequately met, further validating the robustness of our findings. We enclose the normality of residuals test results below for your review, which we also describe in Section 3.6.

Discussion Section:

The negative correlation between direct participation in public health emergencies and psychological resilience deserves deeper exploration. Was this due to trauma exposure, workload, or other factors?

Answer: Thank you for your valuable feedback.

We have expanded on the exploration of the negative correlation between direct participation in public health emergencies and psychological resilience. In our revision, we have discussed public emergencies are often accompanied by high stress and a large number of traumatic scenes. CMATMs face life-and-death decisions, patient suffering and threats to their own safety for long periods of time, which can trigger acute stress reactions or post-traumatic stress disorder (PTSD). Repeated exposure to other people's traumatic experiences can also lead to vicarious trauma, causing emotional numbness, anxiety or depression, and weakening psychological resilience. In addition, we also provided a number of interventions to enrich our discussions.

Suggestions for interventions, such as mindfulness strategies, are useful but could be made more specific and actionable.

Answer: Thank you for your insightful suggestion.

We have already incorporated specific and actionable mindfulness interventions into the revised manuscript. Additionally, we have revisited the discussion section and provided a more detailed description of the proposed intervention measures to enhance the clarity and applicability of the interventions.

The limitations are honestly presented but could be expanded to discuss potential response biases (e.g., social desirability bias in self-reported measures).

Answer: Thank you for your valuable comments.

We have revised the limitations based on your suggestions

We have expanded the limitations section according to your feedback. Our scales are self-report scales, which can lead to potential response bias, such as social desirability bias in self-reported measures. In the future, peer-report scales can be further developed to evaluate the psychological state of CMATMs.

I suggest enriching the discussion by addressing related studies. For example, I recommend the following article:

https://journals.lww.com/jnmr/fulltext/2024/29030/relationship_between_resilience_and_social_and.12.aspx?context=latestarticles

Answer: Thank you for your attention to related studies.

We have already included the recommended article in the revised manuscript and cited it as reference 39 to enhancing the depth of the discussion.

Conclusion:

The conclusion reiterates key findings but lacks actionable recommendations for policymakers or healthcare administrators. Consider framing specific strategies based on identified factors (e.g., training modules, resilie

---

## [Decision Letter · Decision Letter 1]

Dear Dr. Liang,

Thank you for submitting your manuscript to PLOS ONE. After careful consideration, we feel that it has merit but does not fully meet PLOS ONE’s publication criteria as it currently stands. Therefore, we invite you to submit a revised version of the manuscript that addresses the points raised during the review process.

We look forward to receiving your revised manuscript.

Kind regards,

Prof. Ebtsam Abou Hashish,

Academic Editor

PLOS ONE

Journal Requirements:

Additional Editor Comments :

Dear Authors,

Thank you for your revised manuscript and comprehensive responses to the reviewers’ comments. You have addressed many important points with clarity and care. In addition to the reviewers’ comments, please consider the following detailed suggestions to further improve the quality, clarity, and impact of your work:

1. Title and Abstract

Title: The title is clear and descriptive. You may consider adding “in a cross-sectional study” at the end for greater methodological clarity.

Abstract:

You’ve added important methodological details. Still, the conclusion should present more direct applications. For instance, “help them establish a more adaptive defense mechanism” could be made more actionable by suggesting how or through what mechanism (e.g., targeted interventions, training programs).

Reduce redundancy in wording and ensure all key findings are reported with clarity and brevity.

2. Introduction

The revised introduction has improved coherence. However:

Clarify why psychological resilience was prioritized over other constructs such as coping or burnout—e.g., resilience may offer a broader protective framework with intervention potential.

In lines 85–87, when stating the lack of studies on CMATMs’ resilience, consider framing this as a critical research gap given their frontline role in China's foreign health aid policy.

Consider briefly mentioning a theoretical foundation (e.g., conservation of resources theory or stress-appraisal models) to ground your variable selection.

3. Methods

Sample Size Justification: Clearly explain how the 35 explanatory variables were identified, and clarify how many were retained in regression to avoid overfitting concerns.

Inclusion of Non-Medical Staff: Well-justified. Consider reporting the medical vs. non-medical distribution (e.g., n or %) clearly in the results section for transparency.

Study Tools:

Cronbach’s alpha is reported well. Add one sentence clarifying whether any confirmatory factor analysis or cultural validation (beyond translation) was performed or cited.

Statistical Analysis:

You explain the use of parametric vs. non-parametric tests well. Still, please report effect sizes (e.g., η², Cohen’s d, or r) with significance values in key results to improve interpretability.

4. Results

Define all abbreviations directly in each table’s footnotes, even if introduced earlier.

Consider reordering variables in Tables 1 and 2 for better logical flow (e.g., demographics before contextual or behavioral factors).

Highlight in the narrative how many participants fell into each resilience classification, as these are reported numerically in Table 5 but not elaborated upon in-text.

For Table 6 (Multivariate Model), include the adjusted R² value to give readers a sense of overall model strength.

5. Discussion

The link between direct emergency response and reduced resilience is very important. Expand this further by discussing potential mechanisms like trauma exposure, decision fatigue, or moral injury.

The association with professional title could be explained through concepts such as job autonomy, leadership buffering, or perceived control.

When discussing mindfulness interventions, support the suggestion by citing studies that have shown measurable impacts of such strategies on aid workers or healthcare professionals.

Consider providing clearer links between findings and proposed interventions, e.g., how family support could be mobilized through organizational communication strategies or flexible leave policies.

The policy relevance of your findings could be strengthened by summarizing what institutions or agencies could do in practical terms (e.g., onboarding protocols, resilience screening, psychosocial briefings).

6. Conclusion

The conclusion is now more structured. Consider presenting it in three parts:

A summary of the key findings;

A clear set of implications for practice;

A brief roadmap for future studies (e.g., “multi-country comparative studies using longitudinal methods”).

7. Ethical Considerations and Transparency

You provide good detail. Consider specifying whether the online platform used for data collection complied with national data privacy regulations (e.g., China’s Cybersecurity Law).

8. Language and Style

The manuscript is generally well-written, but consider a final grammar check focusing on:

Article use ("a large number of trauma scenes");

Verb tense consistency in results;

Concise language in long compound sentences;

Replacing “strong sense of loneliness” with “pronounced sense of loneliness” as previously suggested.

In several sections, repeated use of “CMATMs” could be replaced with “participants” or “team members” once the term is clear.

Final Recommendation

Your study addresses an important gap in global health resilience research. The manuscript is significantly improved and offers valuable insights. With the above refinements—especially in the discussion and stylistic clarity—it will be ready for publication.

Sincerely,

Prof Ebtsam

Academic Editor

PLOS ONE

Reviewers' comments:

Reviewer's Responses to Questions

**Comments to the Author**

Reviewer #3: All comments have been addressed

2. Is the manuscript technically sound, and do the data support the conclusions?

Reviewer #3: Yes

3. Has the statistical analysis been performed appropriately and rigorously?

Reviewer #3: Yes

4. Have the authors made all data underlying the findings in their manuscript fully available?

Reviewer #3: Yes

5. Is the manuscript presented in an intelligible fashion and written in standard English?

Reviewer #3: Yes

Reviewer #3: The manuscript was well written. All comment from the reviewers (from the previous round) had been addressed. Following are some additional feedback to further improve the manuscript:

Methodology:

Please describe the study design of this study

Line 107: Please check the spelling for form

Line 187: Please provide the full name for the company for excel

Results:

Please rename Table 5 to "The level of Self Efficacy, Anxiety, and Depression on Psychological Resilience

Please do English proof reading to the manuscript.

**Do you want your identity to be public for this peer review?** For information about this choice, including consent withdrawal, please see our Privacy Policy

Reviewer #3: No

---

## [Author Response · Author response to Decision Letter 2]

21 Apr 2025

1. Title and Abstract

Title: The title is clear and descriptive. You may consider adding “in a cross-sectional study” at the end for greater methodological clarity.

Answer: Thank you for your suggestion.

As suggested, we have revised the title to: “Status and Associated Factors of Psychological Resilience of Chinese Medical Aid Team Members under Public Health Emergencies in A Cross-sectional Study”.

Abstract:

You’ve added important methodological details. Still, the conclusion should present more direct applications. For instance, “help them establish a more adaptive defense mechanism” could be made more actionable by suggesting how or through what mechanism (e.g., targeted interventions, training programs).

Answer: Thank you for your insightful comments.

We have revised the conclusion to include more concrete recommendations based on our results. We now propose that interventions could include: regular assessment of CMATMs’ mental health status, strengthening mental health training programs, mobilizing family and social support networks, and improving infrastructure and welfare security policies.

Reduce redundancy in wording and ensure all key findings are reported with clarity and brevity.

Answer: Thank you for valuable feedback.

We have removed unnecessary phrases and rephrased sentences to be more concise. We have also ensured that the reporting of our key findings is direct and to the point, avoiding any ambiguity.

2. Introduction

The revised introduction has improved coherence. However:

Clarify why psychological resilience was prioritized over other constructs such as coping or burnout—e.g., resilience may offer a broader protective framework with intervention potential.

Answer: Thank you for your feedback on the introduction.

We have expanded the introduction to explicitly address this point. We now emphasize that psychological resilience provides a broader, more comprehensive framework for understanding and promoting positive adaptation in the face of adversity. Resilience focuses on how individuals actively adapt to adversity and even grow from the experience, offering a more holistic perspective on mental well-being that goes beyond merely mitigating the negative impacts of stress.

Furthermore, we highlight that focusing on resilience can help us understand the psychological adaptation process of medical aid workers more comprehensively and offer a broader protective framework with intervention potential. We believe this is particularly relevant for medical aid team members who face a wide range of challenges and require a proactive approach to maintaining their psychological health.

In lines 85–87, when stating the lack of studies on CMATMs’ resilience, consider framing this as a critical research gap given their frontline role in China's foreign health aid policy.

Answer: Thank you for highlighting the importance of emphasizing the research gap.

We have revised lines 85-87 to explicitly state this critical research gap. We now emphasize that the limited research on CMATMs’ psychological resilience represents a significant oversight, considering their pivotal role in implementing China’s foreign health aid policy and their exposure to unique stressors in challenging environments. By highlighting this gap, we aim to underscore the urgency and relevance of our study in addressing the psychological well-being of these vital healthcare professionals.

Consider briefly mentioning a theoretical foundation (e.g., conservation of resources theory or stress-appraisal models) to ground your variable selection.

Answer: Thank you for your valuable feedback.

We have added a brief mention of the stress-coping theory to the introduction. We now explain that our study revealed the level of psychological resilience of CMATMs and its related factors based on the stress-coping theory.

3. Methods

Sample Size Justification: Clearly explain how the 35 explanatory variables were identified, and clarify how many were retained in regression to avoid overfitting concerns.

Answer: Thank you for your valuable feedback.

We have carefully reviewed our analysis and consulted with a statistical expert to address these points. We would also like to sincerely apologize for the previous misstatement regarding the number of explanatory variables. Upon closer review and verification, we confirm that there are 34, not 35, explanatory variables in our dataset. We regret any confusion this may have caused and appreciate the reviewer’s attention to detail.

Explanatory variables are also known as independent variables. For the sake of easier understanding, we uniformly refer to them as independent variables.

The 34 independent variables comprised 31 demographic characteristics (detailed in Section 2.2.1, ‘Demographic Information Questionnaire’ which includes two parts with 11 and 20 variables, respectively) and the total scores from three validated scales (Self-Efficacy Scale, Anxiety Scale, and Depression Scale). Furthermore, we also supplemented how variables were identified in Section 2.2.1, ‘Demographic Information Questionnaire’. According to the purpose of the study, research team reviewed previous relevant literature and in combination with the special working environment and living conditions of medical aid team members, formulated a demographic information questionnaire based on the stress-coping theory.

31 demographic variables: The first part included 11 variables, namely age, sex, marital status, highest level of education education, occupation, level of the dispatching organization, professional title, prior overseas experience before participating in foreign aid work, classification of hardship areas in recipient countries, family support, and family fertility.

The second part included 20 variables, namely satisfaction with family life, satisfaction with work, smoking status, frequency of drinking, loneliness, culture and entertainment, frequency of exercise, average daily Internet use time, economic subsidy policy, professional title promotion policy, material condition, political situation, social security situation, language adaptation situation and climate adaptation situation of the recipient country, history of mosquito or insect bites, history of typhoid or malaria or COVID-19 infection, and direct participation in public health emergency response missions in recipient countries.

The demographic variables with statistical significance in the univariate analysis and the total scores of the three scales were taken as independent variables for multiple linear regression. Ultimately, 14 demographic variables and the total scores of three scales were included in the regression analysis.

All the above revisions and clarifications have been supplemented in "2.1 Study Design and participants" and "2.2.1 Demographic information questionnaire".

Inclusion of Non-Medical Staff: Well-justified. Consider reporting the medical vs. non-medical distribution (e.g., n or %) clearly in the results section for transparency.

Answer: Thank you for the suggestion.

We have now added the distribution of medical and non-medical staff in the results section 3.1. The vast majority were medical staff (89.9%), and non-medical staff accounted for 10.1%.

Study Tools:

Cronbach’s alpha is reported well. Add one sentence clarifying whether any confirmatory factor analysis or cultural validation (beyond translation) was performed or cited.

Answer: Thank you for your valuable feedback.

We have added a sentence to the methods section to clarify the validation information for the four scales used in this study.

For example, The Chinese version of the CD-RISC had undergone cross-cultural debugging, confirmatory factor analysis and revision, modifying the five dimensions of the original scale to three dimensions. Reliability, structural validity and predictive validity tests were conducted in the context of Chinese culture to verify the validity and reliability of the Chinese version of GSES. For the SAS, through exploratory factor and confirmatory factor analysis, the factors presented in different populations were stable. SDS was proved to be applicable to the study of depression in Chinese population through the exploratory factor and confirmatory factor analysis

Statistical Analysis:

You explain the use of parametric vs. non-parametric tests well. Still, please report effect sizes (e.g., η², Cohen’s d, or r) with significance values in key results to improve interpretability.

Answer: Thank you for your valuable feedback.

As suggested, to evaluate the strength of the correlation, we calculated the Pearson correlation coefficient (r) as the effect size. we have now added r value in Section 2.4). The value range of r is from -1 to +1. The larger the absolute value, the stronger the correlation.

Additionally, in the results of multiple linear regression, we also reported the adjusted R².

4. Results

Define all abbreviations directly in each table’s footnotes, even if introduced earlier.

Answer: Thank you for your reminder.

We have revised all abbreviations directly in each table’s footnotes, even if introduced earlier.

Consider reordering variables in Tables 1 and 2 for better logical flow (e.g., demographics before contextual or behavioral factors).

Answer: Thank you for your suggestion.

We have reordered the variables in Tables 1 and 2 to present the demographic characteristics before the contextual or behavioral factors.

Highlight in the narrative how many participants fell into each resilience classification, as these are reported numerically in Table 5 but not elaborated upon in-text.

Answer: Thank you for pointing out this.

We have now added a sentence in the section “3.5 The influence of different degree of psychological status on psychological resilience” to explicitly state the number of participants falling into each resilience classification, as presented in Table 5.

The results showed that the proportions of those with excellent, good, average and poor psychological resilience were 30.3%, 23.6%, 25.8, 20.2, respectively.

For Table 6 (Multivariate Model), include the adjusted R² value to give readers a sense of overall model strength.

Answer: Thank you for your careful feedback.

We carefully considered your suggestion and, after further discussion, we believe that incorporating the adjusted R² value into the narrative of Section 3.6, rather than directly into Table 6, is a more appropriate approach in this context.

Therefore, we have added a sentence to Section 3.6 stating that “The adjusted R² for the multivariate model was 0.850, indicating that the model explained approximately 85.0% of the variation degree of psychological resilience, and the model had a relatively high goodness of fit.”

5. Discussion

The link between direct emergency response and reduced resilience is very important. Expand this further by discussing potential mechanisms like trauma exposure, decision fatigue, or moral injury.

Answer: Thank you for highlighting the importance.

We have expanded the discussion section to explore these potential mechanisms in more detail. Public emergencies are inherently characterized by high-stress environments compounded by a multitude of trauma scenes, including mass casualties, severe illness, and the suffering and death of patients. Team members directly involved in the response face life-and-death decisions, patient suffering and threats to their own safety for long periods of time, which can trigger acute stress reactions or post-traumatic stress disorder (PTSD). Repeated exposure to other people's traumatic experiences can also lead to vicarious trauma, causing emotional numbness, anxiety or depression, and weakening psychological resilience. Unlike routine medical practice, these situations often involve unpredictable circumstances, such as resource scarcity, ethical dilemmas surrounding patient prioritization, and the potential for exposure to highly contagious pathogens. Rescue workers need to make a large number of decisions in a short period of time, such as determining the priority order for treating the injured and choosing the best rescue plan, etc. Long-term and high-intensity decision-making may lead to the depletion of cognitive resources, reduce an individual's self-control ability and the ability to cope with stress, thereby affecting psychological resilience.

The association with professional title could be explained through concepts such as job autonomy, leadership buffering, or perceived control.

Answer: Thank you for your valuable feedback.

We have expanded the discussion section to incorporate these ideas. We now discuss how individuals with higher professional titles may experience greater job autonomy, allowing them more control over their work and reducing stress. We also explore the possibility that these individuals may benefit from leadership buffering, where supervisors provide support and protect them from excessive demands.

When discussing mindfulness interventions, support the suggestion by citing studies that have shown measurable impacts of such strategies on aid workers or healthcare professionals.

Answer: Thank you for your valuable comments.

We have now added citations to relevant studies that have shown measurable impacts of mindfulness interventions on aid workers or healthcare professionals in the discussion section. Specifically, we have cited studies that demonstrate the effectiveness of mindfulness-based interventions in reducing burnout and enhancing the resilience of healthcare professionals

Consider providing clearer links between findings and proposed interventions, e.g., how family support could be mobilized through organizational communication strategies or flexible leave policies.

Answer: Thank you for your valuable feedback.

We have expanded the discussion section to provide more specific examples of how the proposed interventions can be implemented to address the identified needs. For instance, we now stated that relevant institutions should establish smooth information channels, organize regular family symposiums, set up family support hotlines, provide psychological counseling and support services for team members and their families, eliminating their doubts and concerns. At the same time, the dispatch and rotation mechanism of foreign aid medical team members should be improved, and personalized dispatch plans can be provided for a period of time. For example, the family visit leave for team members can be appropriately increased, or they can be allowed to return to their home country for leave in advance after completing a certain amount of work.

The policy relevance of your findings could be strengthened by summarizing what institutions or agencies could do in practical terms (e.g., onboarding protocols, resilience screening, psychosocial briefings).

Answer: Thank you for your valuable feedback.

We have revised discussion section by summarizing what institutions or agencies could do in practical terms. We further expanded the discussion around aspects such as onboarding protocols, psychological condition screening and assessment.

6. Conclusion

The conclusion is now more structured. Consider presenting it in three parts:

A summary of the key findings;

Answer: Thank you for your valuable feedback.

We have added a summary of the key findings. This study revealed that CMATMs exhibited lower psychological resilience when characterized by a low sense of self-efficacy, a lower professional title, direct involvement in emergency public health rescue efforts, pronounced sense of loneliness, and limited access to cultural and recreational activities in the recipient area.

A clear set of implications for practice;

Answer: Thank you for your valuable feedback.

We have added a clear set of implications for practice. This study provides guidance for the relevant institutions to select appropriate intervention measures according to the influence factors, which can help foreign aid medical team members enhance their ability to cope with stress, maintain a good working state and efficiency, and thereby enhance the cohesion and collaboration ability of the foreign aid team.

A b

---

## [Editor Report · Decision Letter 2]

Status and Associated Factors of Psychological Resilience of Chinese Medical Aid Team Members under Public Health Emergencies in A Cross-sectional Study

PONE-D-25-03434R2

Dear Dr. Yinhua Liang

We’re pleased to inform you that your manuscript has been judged scientifically suitable for publication and will be formally accepted for publication once it meets all outstanding technical requirements.

Kind regards,

Prof. Ebtsam Aly Abou Hashish,

Academic Editor

PLOS ONE

Additional Editor Comments (optional):

The revised manuscript titled “Status and Associated Factors of Psychological Resilience of Chinese Medical Aid Team Members under Public Health Emergencies in A Cross-sectional Study” has been significantly improved and now meets the journal’s standards for publication.

The authors have addressed all reviewer concerns, clarified methodological details, strengthened theoretical and practical implications, and improved the overall structure and language of the manuscript. No substantive issues remain that would require further revision.

I recommend acceptance of the manuscript in its current form.

Reviewers' comments:

-

<gdiv id="ginger-floatingG-container" style="position: absolute; top: 0px; left: 0px;"><gdiv class="ginger-floatingG ginger-floatingG-closed ginger-floatingG-posdown" style="display: block; left: 651.5px; top: 157.891px; z-index: 51;"><gdiv class="ginger-floatingG-disabled-main"><gdiv class="ginger-floatingG-bar-tool-tooltip ginger-floatingG-bar-tool-tooltip-enable">Enable Ginger</gdiv></gdiv><gdiv class="ginger-floatingG-offline-main"><gdiv class="ginger-floatingG-bar-tool-tooltip">*Cannot connect to Ginger* Check your internet connection

Go Premium

---

## [Editor Report · Acceptance letter]

PONE-D-25-03434R2

PLOS ONE

Dear Dr. Liang,

I'm pleased to inform you that your manuscript has been deemed suitable for publication in PLOS ONE. Congratulations! Your manuscript is now being handed over to our production team.

Kind regards,

on behalf of

Prof Ebtsam Aly Abou Hashish

Academic Editor

PLOS ONE